# Lifting integrable models and long-range interactions

**Marius de Leeuw and Ana Lucia Retore**

Hamilton Mathematics Institute, School of Mathematics,
Trinity College Dublin, Dublin, Ireland

## Abstract

In this paper we discuss a constructive approach to check whether a constant Hamiltonian is Yang-Baxter integrable. We then apply our method to long-range interactions and find the Lax operator and $R$-matrix of the three-loop SU(2) sector in N=4 SYM. We show that all known integrable long-range deformations of the 6-vertex models of this type can be obtained from a Lax operator and an $R$-matrix. Finally we discuss what happens at higher loops and highlight some general structures that these models exhibit.

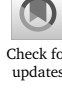
# 1 Introduction

The question whether a model is integrable or not is an important one. Integrable models have special behaviour and allow for a whole range of techniques that can be used to obtain exact results. Most of these techniques, such as the algebraic Bethe Ansatz, are based on the existence of a Lax-matrix and an $R$-matrix that are subject to certain relations.

Recently, we have started a research direction that focuses on the classification of regular solutions of the quantum Yang-Baxter equation [1–5]. However, from a practical point of view, using our classification to decide whether a Hamiltonian is integrable and descends from an $R$-matrix is not always very practical. First, there is a lot of freedom in possible identifications and dependence on the spectral parameter. Second, the model under consideration would have to be in the set of models that have been classified so far and this is not always obvious.

In this paper we try to fill this gap by proposing a constructive method to derive the Lax operator and $R$-matrix for a Hamiltonian and show whether or not it comes from a regular integrable model. We will demonstrate the method by looking at 6-vertex models, where the situation is under control and very well-understood.

Next we focus on a set of integrable spin chains where the existence of a Lax operator and $R$-matrices are still an open problem. These are the perturbative spin chains that appear in the context of AdS/CFT, see [6] for a review. At one loop, the dilatation operator can be mapped to an integrable nearest neighbour spin chain [7]. However, at each increasing order in perturbation theory, the interaction range of the corresponding spin chain increases [8–10]. Hence, the spin chain Hamiltonian takes the form

$$\mathcal{H} = \mathcal{H}_{NN} + g^2 \mathcal{H}_{NNN} + \dots \tag{1}$$

At each order in $g$ these spin chains are perturbatively integrable and the full Hamiltonian would have infinite range. We say that a spin chain is perturbatively integrable if all relations related to integrability (like the commutation of the conserved charges, the Yang-Baxter equation and the fundamental commutation relations) are satisfied *up to the order* we specify.

A general frame work for these types of spin chains was put forward in [11, 12]. The idea is to introduce a long-range deformation of a nearest neighbour spin chain in a perturbative way. However, this formalism focuses on the level of the charges. Indeed, the idea is to perturbatively introduce long-range deformations of the charges of the system in such a way that they still commute. However, it is unclear if these charges come from a Lax operator and if there is a corresponding solution of the Yang-Baxter equation. In this paper we will show that seems to be the case and we will derive the Lax operator and the $R$-matrix for the two-loop Hamiltonian in the SU(2) sector of $\mathcal{N} = 4$ SYM.

To this end, we will build on the formalism of medium range spin chains studied in [13] and [14]. The idea is to double the local Hilbert space so that the interaction range gets increased. We will find additional evidence for some of the conjectures put forward in [14] in this context.

In [15] the perturbative long-range deformations for a general 6-vertex model were classified and could be mapped one-to-one to solutions of the so-called deformation equation introduced in [11, 12]. Here we will repeat this classification but starting from our method and we find that they coincide. This indicates that perturbative long-range deformations are all descendant from a Lax operator and a solution of the Yang-Baxter equation. We then go to three loops and we furthermore find that the Lax operator and the $R$-matrix satisfy some interesting properties. In particular, we find that our Lax operator factorises as conjectured in [14] and reduces to a Lax operator with a larger auxiliary space.

This paper is organised as follows. We first give an overview of regular integrable spin chains and the equations that Lax operators and $R$-matrices satisfy. Then we discuss how to

check for a given Hamiltonian whether it is integrable or not and how to find the corresponding Lax operator. We demonstrate our method on a 6-vertex model. After this we turn our attention to long-range spin chains and focus in particular on the $SU(2)$ sector in $\mathcal{N} = 4$ super Yang–Mills theory. We classify long-range deformations of 6-vertex models and finally discuss some further observations and applications of our results. We end with conclusions and a discussion.

**Note added**   During the final stages of this paper we became aware of [16], which has overlap with this work.

## 2   Regular integrable spin chains

Let us define what we mean by a regular Yang-Baxter integrable spin chain and discuss some of its properties.

**Definition**   We consider a homogeneous spin chain of length $L$ with local Hilbert space $V$ and Hamiltonian $H$. Suppose there is a Lax operator $\mathcal{L}$ and an $R$-matrix $R$

$$\mathcal{L}(u) : V \otimes V \to V \otimes V, \qquad\qquad R(u, v) : V \otimes V \to V \otimes V, \qquad (2)$$

such that $\mathcal{L}$ satisfies the fundamental commutation or RLL relations

$$R_{12}(u, v)\mathcal{L}_{1a}(u)\mathcal{L}_{2a}(v) = \mathcal{L}_{2a}(v)\mathcal{L}_{1a}(u)R_{12}(u, v), \qquad (3)$$

and $R$ satisfies the quantum Yang-Baxter equation

$$R_{12}(u_1, u_2)R_{13}(u_1, u_3)R_{23}(u_2, u_3) = R_{23}(u_2, u_3)R_{13}(u_1, u_3)R_{12}(u_1, u_2). \qquad (4)$$

We call the spin chain (Yang-Baxter) integrable if the Hamiltonian can be written as the logarithmic derivative of the transfer matrix $t(u)$

$$H = \frac{d}{du} \log t(u)\Big|_{u=0}, \qquad\qquad t(u) \equiv \mathrm{tr}_a[\mathcal{L}_{aL}(u)\dots\mathcal{L}_{a1}(u)]. \qquad (5)$$

Along the paper we will call this type of Hamiltonian constant, since it does not depend on the spectral parameter $u$. We call the spin chain (5) regular if

$$\mathcal{L}_{ia}(0) = P_{ia}, \qquad (6)$$

where $P_{ia}$ is the permutation operator acting on sites $i$ and $a$. From this it follows that the $R$-matrix is regular in the sense that

$$R_{ab}(u, u) = P_{ab}. \qquad (7)$$

This can be shown by considering (3) at the point $v = u$. At this point we get

$$R_{12}(u, u)\mathcal{L}_{1a}(u)\mathcal{L}_{2a}(u) = \mathcal{L}_{2a}(u)\mathcal{L}_{1a}(u)R_{12}(u, u). \qquad (8)$$

We can expand this around $u = 0$ and use the regularity of $\mathcal{L}$. To leading order it is clear we need $R(0,0) = P$. Let us then write $R(u,u) = P(1 + Au + \dots)$. Plugging this into the above equation and expanding to first order implies that $A \sim 1$ and hence it follows by induction that $R(u,u)$ is proportional to the permutation operator and is regular.

   A special solution of the fundamental commutation relations is the case where $\mathcal{L} \sim R$. In this case (3) becomes equivalent to the Yang-Baxter equation. Furthermore notice that we take the auxiliary space to be the same as the local Hilbert space. More generally they can be different, but in that case the regularity condition (6) needs to be modified. We will see an example of this when we discuss long-range interactions.

**Conserved charges**    From the the fundamental commutation relations (3) it is easy to show that

$$[t(u), t(v)] = 0.$$ 
(9)

Hence by expanding the transfer matrix as a power series in $u$, we find that the Hamiltonian is part of a family of conserved, commuting charges.

At the special point $u = 0$ the transfer matrix becomes the shift operator $U$

$$t(0) = \text{tr}_a P_{aL} \dots P_{a1} = P_{12} \dots P_{L-1L} \equiv U.$$ 
(10)

Let us denote the derivative of the Lax operator by a dot, then we obtain

$$\dot{t}(u) = \sum_n \text{tr}_a \Big[ \mathcal{L}_{aL} \dots \dot{\mathcal{L}}_{an} \dots \mathcal{L}_{a1} \Big].$$ 
(11)

We define the Hamiltonian density by $\dot{\mathcal{L}}_{ab}(0) = P_{ab} \mathcal{H}_{ab}$. When we evaluate the logarithmic derivative at $u = 0$

$$\mathbb{Q}_2 \equiv H = t(0)^{-1} \dot{t}(0)$$ 
(12)

$$= U^{-1} \sum_n \text{tr}_a \Big[ P_{aL} \dots P_{an} \mathcal{H}_{an} \dots P_{a1} \Big]$$ 
(13)

$$= \sum_n \mathcal{H}_{nn+1},$$ 
(14)

we obtain a charge that has nearest-neighbour interactions. The next term in the expansion depends on the second derivative of the $R$-matrix $\ddot{\mathcal{L}}_{ab}(0) \equiv P_{ab} \mathcal{A}_{ab}$ and takes the form

$$\mathbb{Q}_3 \equiv t(0)^{-1} \ddot{t}(0) - \mathbb{Q}_2^2$$ 
(15)

$$= \sum_i \mathcal{A}_{i-1i} + 2 \sum_{i>j} \mathcal{H}_{i-1,i} \mathcal{H}_{j-1,j} - \sum_{i,j} \mathcal{H}_{i-1,i} \mathcal{H}_{j-1,j}$$

$$= \sum_i \Big[ \mathcal{A}_{i-1,i} - \mathcal{H}_{i-1,i}^2 \Big] - \sum_i [\mathcal{H}_{i-1,i}, \mathcal{H}_{i,i+1}].$$ 
(16)

Notice that this expansion of the transfer matrix holds in general and does not depend on the fundamental commutation relations. For an integrable spin chain, however, $\mathbb{Q}_2$ and $\mathbb{Q}_3$ need to commute which will put restrictions on $\mathcal{H}$ and $\mathcal{A}$.

**Boost operator**    We can define a boost operator that recursively generates the charges of the spin chain [17]. Let us expand the fundamental commutation relations around $u = v$. Let us denote

$$\tilde{\mathcal{H}}_{ab}(v) = P_{ab} \frac{d}{du} R_{ab}(u, v) \Big|_{u \to v}.$$ 
(17)

Notice that contrary to the definition in (5), this density Hamiltonian is not constant, since it still depends on the spectral parameter $v$. Along the paper we will refer to this type as *functional*.[1] We find that the first order in (3) is automatically satisfied by regularity of $R$ and at the next order we find

$$[\tilde{\mathcal{H}}_{12}, \mathcal{L}_{1a} \mathcal{L}_{2a}] = \dot{\mathcal{L}}_{1a} \mathcal{L}_{2a} - \mathcal{L}_{1a} \dot{\mathcal{L}}_{2a}.$$ 
(18)

---

[1]This distinction between constant and functional Hamiltonian is very important in the context of the boost automorphism mechanism (see for example [3]), where it allows to construct difference-form ($R_{ij}(u_i, u_j) = R_{ij}(u_i - u_j)$) and non-difference form ($R_{ij}(u_i, u_j) \neq R_{ij}(u_i - u_j)$), respectively.

This is the Sutherland equation. Let us put $u \to 0$, then we find

$$\tilde{\mathcal{H}}_{2a}(0) - \tilde{\mathcal{H}}_{12}(0) = \mathcal{H}_{2a} - \mathcal{H}_{12}. \tag{19}$$

Hence, we find that $\tilde{\mathcal{H}}(0) = \mathcal{H}$. Since $R$ is the unique solution of the Sutherland equation, this implies that

$$\mathcal{L}(u) = R(u, 0). \tag{20}$$

Let us consider a spin chain of infinite length. From the Sutherland equation we can then show that

$$\sum_{k \in \mathbb{Z}} k[t(u), \tilde{\mathcal{H}}_{k,k+1}(u)] = \dot{t}(u). \tag{21}$$

Writing $\mathbb{Q}_1 = t(0) = U$ and the remaining charges $\mathbb{Q}_i$ as

$$\log t(u) = \sum_{r=2}^{\infty} \mathbb{Q}_r u^{r-1}, \qquad\qquad \tilde{\mathcal{H}} = \mathcal{H} + \sum_{r=1}^{\infty} \mathcal{H}^{(r)} u^r, \tag{22}$$

we derive that

$$\mathbb{Q}_{r+1} = \sum_{s=0}^{r-1} [\mathcal{B}[\mathcal{H}^{(s)}], \mathbb{Q}_{r-s}], \qquad\qquad \mathcal{B}[\mathcal{M}] = \sum_{k} k \mathcal{M}_{k,k+1}. \tag{23}$$

We note that $\mathcal{H}^{(0)} = \mathcal{H}$. The operator $\mathcal{B}[\mathcal{M}]$ is called the boost operator associated with the operator $\mathcal{M}$. For instance, we see that

$$\mathbb{Q}_3 = -\sum_{k} [\mathcal{H}_{k-1,k}, \mathcal{H}_{k,k+1}] + \mathcal{H}^{(1)}_{k,k+1}. \tag{24}$$

Comparing this against (16), we see that we can identify $\mathcal{A}_{k,k+1} = \mathcal{H}^2_{k,k+1} + \mathcal{H}^{(1)}_{k,k+1}$.

## 3 Lifting a constant Hamiltonian

In previous works we focused on classifying solutions of the Yang-Baxter equation. However, in practice going through the solutions in the classification while accounting for all possible identifications, such as basis transformations, twists and reparameterizations is not the most efficient approach to determine if a spin chain is integrable. In this section we will discuss a structured approach to determine whether a given constant Hamiltonian comes from a regular integrable spin chain.

**Approach** Consider a spin chain with a nearest neighbour Hamiltonain $H$. Let us assume that it comes from a regular integrable spin chain. This means that there is a tower of conserved charges $\mathbb{Q}_r$ with $\mathbb{Q}_2 = H$ that mutually commute

$$[\mathbb{Q}_r, \mathbb{Q}_s] = 0. \tag{25}$$

From the boost operator (24), we find that we can express $\mathbb{Q}_3$ in terms of $\mathcal{H}$ up to an unknown range 2 term $\mathcal{H}^{(1)}$. Hence, imposing that

$$[\mathbb{Q}_2, \mathbb{Q}_3] = 0, \tag{26}$$

will constrain $\mathcal{H}^{(1)}$. Since $\mathbb{Q}_3$ is of range 3, it is easy to see that this generically gives a set of

equations on the components of $\mathcal{H}^{(1)}$ which is overdetermined. Thus, there are two possibilities

1. There is no solution $\Rightarrow$ the spin chain is *not* Yang-Baxter integrable.

2. There is a solution $\Rightarrow$ the spin chain is *potentially* integrable.

As with our classification approach, it seems that in case two, the spin chain is actually always integrable. However, this is only backed by experimental evidence and we have no proof of this at this point.

In the second case we will have solved $\mathcal{H}^{(1)}$ and then we can compute $\mathbb{Q}_4$, which can be seen from (23) to depend on $\mathcal{H}, \mathcal{H}^{(1)}$ and $\mathcal{H}^{(2)}$. Again we find that the only new information for this operator is encoded in a new range 2 term and we can repeat the above process to compute $\mathcal{H}^{(2)}$ and so on. This will allow us to completely reconstruct both $\mathcal{L}$ and $\tilde{\mathcal{H}}$, from which we can easily recover the *R*-matrix and check that both the fundamental commutation relations and the Yang-Baxter equation are satisfied.

**Freedom** Just like in the R-matrix classification, there is a redundancy in the solution space. This corresponds to different ways a constant Hamiltonian can be extended into a functional one. This differs slightly from our previous works. Since the starting point is a fixed Hamiltonian, the normalisation and reparameterisation symmetries are modified. We can take

$$\mathcal{L}(u) \mapsto f(u)\mathcal{L}(g(u)), \tag{27}$$

but we need to have

$$f(0) = 1, \qquad\qquad g(0) = 0, \tag{28}$$
$$f'(0) = 0, \qquad\qquad g'(0) = 1, \tag{29}$$

in order for this identification to be compatible with the boundary conditions. Clearly these degrees of freedom will show up starting from the quadratic order in the expansion of $\mathcal{L}$.

Furthermore, we can consider a basis transformation in the auxiliary space

$$\mathcal{L}_{ia}(u) \mapsto V_a(u)\mathcal{L}_{ia}(u)V_a^{-1}(u). \tag{30}$$

Since the transfer matrix is defined by a trace over the auxiliary space, such maps will drop out of the explicit form of any charge. These identifications can be used to bring $\mathcal{L}$ to a nice form, but when solving the coefficients order by order it is not always clear what the best choice is.

**Practical implementation** In practice, the above approach can be very efficiently implemented on the computer. Since for each operator, the new information is encoded in a range 2 term, we found that it is sufficient to take logarithmic derivatives of the transfer matrix on spin chains of low length. Writing

$$\mathcal{L}(u) = P\left(1 + u\mathcal{H} + \sum_{i>1}\frac{\mathcal{L}^{(i)}u^i}{i!}\right), \tag{31}$$

it is straightforward to explicitly compute the charges $\mathbb{Q}$ on a spin chain of small length and by imposing that they all commute fixes the components $\mathcal{L}^{(i)}$. Because of this, it is feasible to compute charges up to $\mathbb{Q}_{20}$ without too much problems. Consequently, one finds the first 20 terms in the expansion of $\mathcal{L}$. At this point resumming the power series is usually doable and some closed formulas can be obtained.

However, once several components of the Lax operator have been computed, there is a faster way to compute the remaining ones. In particular, on a spin chain of sufficient length, we can compute the transfer matrix with the partially known Lax operator. This operator has to commute with the Hamiltonian. This relates the known and unknown coefficients of $\mathcal{L}$ and usually allows for a very easy way to fix the full Lax operator. Finally, once $\mathcal{L}$ is computed $R$ can be solved from the fundamental commutation relations linearly.

**The 6-vertex B model**   As an example to demonstrate our procedure, let us consider a density Hamiltonian given by

$$\mathcal{H} = h_1 + h_2(\sigma_z \otimes 1 - 1 \otimes \sigma_z) + h_3 \sigma_+ \otimes \sigma_- + h_4 \sigma_- \otimes \sigma_+ + h_5(\sigma_z \otimes 1 + 1 \otimes \sigma_z). \tag{32}$$

and a corresponding Lax operator $\mathcal{L}^{(i)}$ of the form

$$\mathcal{L}^{(i)} = l_1^{(i)} + l_2^{(i)}(\sigma_z \otimes 1 - 1 \otimes \sigma_z) + l_3^{(i)} \sigma_+ \otimes \sigma_- + l_4^{(i)} \sigma_- \otimes \sigma_+ + l_5^{(i)}(\sigma_z \otimes 1 + 1 \otimes \sigma_z) + l_6^{(i)} \sigma_z \otimes \sigma_z$$

$$\equiv \begin{pmatrix} \tilde{l}_1^{(i)} & 0 & 0 & 0 \\ 0 & \tilde{l}_2^{(i)} & \tilde{l}_3^{(i)} & 0 \\ 0 & \tilde{l}_4^{(i)} & \tilde{l}_5^{(i)} & 0 \\ 0 & 0 & 0 & \tilde{l}_6^{(i)} \end{pmatrix}. \tag{33}$$

Substituting (31) and (33) in the transfer matrix we can compute the coefficients in the Lax operator perturbatively. For example, we can use $[\mathbb{Q}_2, \mathbb{Q}_3] = 0$ to determine $\mathcal{L}^{(2)}$, $[\mathbb{Q}_2, \mathbb{Q}_4] = 0$ to determine $\mathcal{L}^{(3)}$ and so on. For simplicity let us consider $h_1 = 0$. Moreover, on a periodic chain the term proportional to $h_2$ vanishes. This agrees with the observation in [3,4] that such terms follow from a local basis transformations, which are irrelevant. Hence, we derive

$$\mathbb{Q}_2 = \sum_{i=1}^{4} \left( h_3 \, \sigma_i^+ \sigma_{i+1}^- + h_4 \, \sigma_i^- \sigma_{i+1}^+ + 2 h_5 \, \sigma_i^z \right), \tag{34}$$

and

$$\begin{aligned}
\mathbb{Q}_3 = {}& -\left( 8h_2^2 + 2h_3 h_4 + 8h_5^2 - 2l_1^{(2)} - l_3^{(2)} - l_4^{(2)} - 2l_6^{(2)} \right) \mathbb{I} + 2l_5^{(2)} \sum_i \sigma_i^z \\
& + \frac{1}{4} \left( 8h_2^2 + 2h_3 h_4 - 8h_5^2 + 2l_1^{(2)} - l_3^{(2)} - l_4^{(2)} + 2l_6^{(2)} \right) \sum_i \sigma_i^z \sigma_{i+1}^z \\
& - \left( 4h_3 h_5 - l_1^{(2)} + 2l_2^{(2)} + l_6^{(2)} \right) \sum_i \sigma_i^+ \sigma_{i+1}^- \\
& + \left( 4h_3 h_5 + l_1^{(2)} + 2l_2^{(2)} - l_6^{(2)} \right) \sum_i \sigma_i^- \sigma_{i+1}^+ \\
& + \sum_i \left( h_3^2 \sigma_i^+ \sigma_{i+2}^- - h_4^2 \sigma_i^- \sigma_{i+2}^+ \right) \sigma_{i+1}^z.
\end{aligned} \tag{35}$$

By requiring $[\mathbb{Q}_2, \mathbb{Q}_3] = 0$ we obtain that

$$\tilde{l}_6^{(2)} = -8h_2^2 - 2h_3 h_4 + 8h_5^2 - \tilde{l}_1^{(2)} + \tilde{l}_3^{(2)} + \tilde{l}_4^{(2)}, \tag{36}$$

and no restrictions on the other $\tilde{l}_i^{(2)}$. By computing $[\mathbb{Q}_2, \mathbb{Q}_4] = 0$ we find again a condition for $\tilde{l}_6^{(3)}$ but the other $\tilde{l}_i^{(3)}$ remain free. We repeated this process until $\mathbb{Q}_8$ and found only one

condition per step. The remaining degrees of freedom have a clear interpretation that we will discuss shortly, but let us now put all the free $\tilde{l}_i^{(j)}$ to zero for simplicity which yields

$$\mathcal{L}(u) = \begin{pmatrix} 1+2uh_5 & 0 & 0 & 0 \\ 0 & uh_4 & 1-2uh_2 & 0 \\ 0 & 1+2uh_2 & uh_3 & 0 \\ 0 & 0 & 0 & 1-2uh_5+u^2\mathcal{A}(u) \end{pmatrix}, \tag{37}$$

where

$$\mathcal{A}(u) = (4h_2^2 + h_3h_4 - 4h_5^2)(-1+2uh_5-4u^2h_5^2+8u^3h_5^3-16u^4h_5^4+32u^5h_5^5+...+). \tag{38}$$

We can see that the expression for $\mathcal{A}(u)$ can be easily summed, such that we can write the complete $\mathcal{L}(u)$ as

$$\mathcal{L}(u) = \begin{pmatrix} 1+2uh_5 & 0 & 0 & 0 \\ 0 & uh_4 & 1-2uh_2 & 0 \\ 0 & 1+2uh_2 & uh_3 & 0 \\ 0 & 0 & 0 & \frac{1-u^2(4h_2^2+h_3h_4)}{1+2uh_5} \end{pmatrix}. \tag{39}$$

In the next step we use the RLL relations to very easily obtain the corresponding R-matrix

$$R(u) = \begin{pmatrix} \frac{u(1-4v^2h_2^2)+(u-v)f(v)}{v(1-2uh_2)(1+2vh_2)} & 0 & 0 & 0 \\ 0 & \frac{(vf(u)-uf(v))h_4}{(1-2uh_2)(1+2vh_2)} & 1 & 0 \\ 0 & \frac{(1+2uh_2)\,(1-2vh_2)}{(1-2uh_2)\,(1+2vh_2)} & \frac{(u-v)h_3}{(1-2uh_2)(1+2vh_2)} & 0 \\ 0 & 0 & 0 & \frac{v(1-4u^2h_2^2)-(u-v)f(u)}{u(1-2uh_2)(1+2vh_2)} \end{pmatrix}, \tag{40}$$

where

$$f(u) = \frac{u^2(4h_2^2+h_3h_4)-1}{1+2uh_5}. \tag{41}$$

Of course this is a case that was already classified in [3,4] and it is illustrative to compare the results. The Hamiltonian (32) is of type 6-vertex B and the corresponding $R$-matrix depends on several free functions. In order to give a regular $R$-matrix with the correct Hamiltonian, these functions will need to specify some boundary conditions, but are otherwise free. If we write the functions as a Taylor series in $u$, then the boundary conditions will fix the lowest two orders, but not the rest. This is exactly what we see in this way of solving for our Lax operator.

## 4 Long-range interactions

A more interesting direction are spin chains with long-range interactions. In this section we will focus on the framework of long-range interactions that was introduced in [11, 12]. The idea is to introduce a long-range deformation of a nearest neighbour spin chain in a perturbative way.

**Framework** Consider a regular integrable spin chain with conserved charges $\{\mathbb{Q}_r^{(0)}\}$. The range of $\mathbb{Q}_r^{(0)}$ is $r$. Let us introduce a coupling constant $g$ and write

$$\mathbb{Q}_r(g) = \mathbb{Q}_r^{(0)} + g\mathbb{Q}_r^{(1)} + g^2\mathbb{Q}_r^{(2)} + \dots \tag{42}$$

We impose that $\mathbb{Q}_r^{(n)}$ is of range $r + n$. This clearly defines a long-range deformation which is integrable if we also impose that

$$[\mathbb{Q}_r(g), \mathbb{Q}_s(g)] = 0. \tag{43}$$

This needs to hold at each order in $g$ and this allows us to define a perturbative notion of long-range interaction, where the spin chain is integrable up to some order in $g$.

In [11,12], and later [15] it is further argued and checked that all long-range deformations correspond to the solutions of the deformation equation

$$\frac{d}{dg}\mathbb{Q}_r(g) = [X(g), \mathbb{Q}_r(g)], \tag{44}$$

where $X(g) = \sum_{n=0}^{\infty} X^{(n)} g^n$. Thus, we obtain the perturbative solution

$$\mathbb{Q}_r^{(n+1)} = \sum_{m=0}^{n} [X^{(m)}, \mathbb{Q}_r^{(n-m)}]. \tag{45}$$

There are different types of operators $X$ which give an integrable deformation with increasing range, namely

- local operators,

- boosted charges,

- bilocal charges.

Both the boosted charges and the bilocal operators are only defined on an infinite open chain, but the commutator on the RHS of the deformation equation will be a finite range operator that can consistently be restricted to a spin chain of finite length. A fourth possibility is a basis transformation which simply mixes the conserved charges of the original chain

$$\mathbb{Q}_r \to \sum_s \gamma_{r,s}(g)\mathbb{Q}_s. \tag{46}$$

This is of course a trivial deformation as it has no effect on the eigenvectors and acts trivially on the eigenvalues of the spin chain.

**Long-range Lax operator**    However, the approach from [11,12] focuses on the charges and it is not clear if there is a Lax operator and $R$-matrix that underlie these deformations. In order to include long-range interactions we need to increase the size of our Hilbert space and consider a spin-ladder system. In the context of medium-range interactions, this was recently applied in [14].

Consider a spin chain with local Hilbert space $V$ in which the Hamiltonian $\mathcal{H}_V$ is of range 3. If we assume the spin chain has even length, then we can define a new spin chain with local space $W = V \otimes V$. On this spin chain, the Hamiltonian $\mathcal{H}_W$ which corresponds to $\mathcal{H}_V$ lifted to the larger spin chain is now nearest neighbour and we apply our regular spin chain formalism again. This can easily be generalised to longer range interactions.

For simplicity let us restrict to range 3. Hence, we look for a Lax operator of the form $\mathcal{L}_{a_1, a_2, n_1, n_2}$, acting on $W \otimes W$ and the indices $a_i, n_i$ reflect the decomposition of $W = V \otimes V$. In [14] it was argued and conjectured that such a Lax operator should actually split into two factors

$$\mathcal{L}_{a_1, a_2, n_1, n_2}(u) = \mathcal{L}_{a_1, a_2, n_2}(u)\mathcal{L}_{a_1, a_2, n_1}(u). \tag{47}$$

In all the examples we have worked out and will present in the following sections, we found that this was indeed the case. Hence it seems like all the perturbative long-range models of range $r$ can be build from a Lax operator whose auxiliary space is $V^{\otimes(r-1)}$ and whose physical space is unchanged. Furthermore, in [14] the corresponding regularity condition was also derived and is

$$\mathcal{L}_{a_1,\dots a_n,i}(0) = P_{a_1,i}\dots P_{a_n,i} = P_{i,a_n}P_{a_n,a_{n-1}}\dots P_{a_2,a_1}. \tag{48}$$

Because of this we conjecture, analogous to [14], that we can derive the Lax operator for perturbative long-range interactions by applying the formalism from Section 3 with such a Lax operator instead. This will mean a significant computational simplification.

## 5 Two-loop SU(2) sector

As a first application let us consider the case where our Lax operator is $SU(2)$ invariant. This long-range spin chain corresponds to the $SU(2)$ sector in $\mathcal{N}=4$ SYM. The interaction range of the Hamiltonian grows with every loop order and to three loops it is given by [7–9, 18]

$$\begin{aligned}
H = {} & \{\} - \{1\} + g^2\left(-2\{\} + 3\{1\} - \frac{1}{2}(\{1,2\} + \{2,1\})\right) \\
& + g^4\left(\frac{15}{2}\{\} - 13\{1\} + \frac{1}{2}\{1,3\} + 3(\{1,2\} + \{2,1\}) - \frac{1}{2}(\{1,2,3\} + \{3,2,1\})\right) + \mathcal{O}\left(g^5\right),
\end{aligned} \tag{49}$$

where

$$\{p_1, p_2, \dots\} = \sum_{p=1}^{L} P_{p+p_1, p+p_1+1}P_{p+p_2, p+p_2+1}\cdots \tag{50}$$

As far as integrability is concerned, we can write the density Hamiltonian $\mathcal{H}$ corresponding to (49) as given by

$$\mathcal{H} \sim 1 - P_{12} - g^2 P_{13} + g^4 P_{14} + \dots, \tag{51}$$

by using identities like $P_{12}P_{23} + P_{23}P_{12} = P_{12} + P_{23} + P_{13} - \mathbb{I}$ and changing normalization and adding a constant shift. Given that we can easily put those terms back in the Lax operators, for simplicity we will continue working with the simple expression (51).

Even though integrability was shown in [9], the approach focuses on the charges and not on the $R$-matrix. In this section we will show that this model is Yang-Baxter integrable and compute the corresponding $R$-matrix and Lax operator that generates them.

**Doubling XXX**  Let us first have a look at which range 3 interactions are allowed. Let us start with the XXX spin chain Hamiltonian

$$\mathcal{H}_{12} = 1 - P_{12}. \tag{52}$$

It is generated by the usual Lax operator and $R$-matrix

$$\mathcal{L}_{ai}(u) = \frac{u\,1 - P_{ai}}{u-1}, \qquad\qquad R_{ab}(u) = \frac{u\,1 - P_{ab}}{u-1}. \tag{53}$$

In order to allow for range 3 interactions, we need to consider an auxiliary space of double dimension and a bigger corresponding Lax operator $\mathcal{L}^{(3)}_{a_1 a_2 i}$.

Hence, starting from the XXX spin chain Hamiltonian, we can lift this Hamiltonian to a Lax operator of the form $\mathcal{L}^{(3)}_{a_1 a_2 i}$ by applying our formalism. We find that the most general solution is

$$\mathcal{L}^{(3)}_{a_1 a_2 i}(u) = A(u)V_{a_1 a_2}(u)\mathcal{L}_{a_1 i}(f(u))\mathcal{L}_{a_2 i}(g(u))V^{-1}_{a_1 a_2}(u), \tag{54}$$

where $A$ is an overall normalisation, $f(u), g(u)$ are smooth functions such that

$$f(0) = g(0) = 0, \qquad\qquad f'(0) + g'(0) = 1, \qquad (55)$$

and

$$V = \alpha(u)1 - P, \qquad (56)$$

is a basis transformation in the auxiliary space. Clearly such a basis transformation will drop out of the transfer matrix due to cyclicity of the trace. It is easy to check that $\mathcal{L}^{(3)}$ satisfies the correct boundary conditions and satisfies the usual RLL relations with the following $R$-matrix

$$R_{a_1 a_2, b_1 b_2}(u, v) = R_{a_1, b_2}(f(u), g(v))R_{a_1, b_1}(f(u), f(v))R_{a_2, b_2}(g(u), g(v))R_{a_2, b_1}(g(u), f(v)). \qquad (57)$$

This is of course unsurprising since this corresponds to the well-known way of doubling a spin chain and simply expresses the $R$-matrix as pairwise scattering of four particles

$$(12)(34) \rightarrow (34)(12). \qquad (58)$$

Hence at this level, the Lax operator and $R$-matrix simply decomposes into fundamental ones.

**Long-range deformations of XXX**   Let us now add a range 3 part. It is easy to check that there are only two range 3 interactions that are compatible with $SU(2)$ symmetry. They are

$$\mathcal{O}_1 = P_{13}, \qquad\qquad \mathcal{O}_2 = [P_{12}, P_{23}]. \qquad (59)$$

The operator $\mathcal{O}_2$ corresponds to the third charge of the XXX spin chain and hence it can be added to the Hamiltonian as a trivial deformation as it will simply commute with $\mathcal{H}$. We will discard it for that reason and consider

$$\mathcal{H}_{123} = 1 - P_{12} - g^2 P_{13}, \qquad (60)$$

corresponding to (51). On periodic chains we only need to add $P_{12}$ and can discard possible $P_{23}$ terms.

We find that this Hamiltonian is indeed Yang-Baxter integrable and we are able to construct the Lax operator and $R$-matrix. Remarkably, we find that adding the $g^2$ part influences the $g^0$ part from (54). More precisely, it fixes $g(u) = 0$. After fixing our remaining freedom such that $V = 1$ and $f = u$, we find

$$\mathcal{L}_{a_1 a_2 i}(u) = P_{a_1 i} P_{a_2 i} \left[ P_{a_1 a_2} \mathcal{L}_{a_1 a_2}(u) - \frac{2g^2 u}{(u-2)(u^2-1)} P_{a_1 i} \right]. \qquad (61)$$

A second remarkable feature is that this Lax operator only describes an integrable spin chain at order $g^2$. *There is no range 3 extension of this Lax operator that works at order $g^4$.* Hence in order to make this spin chain fully integrable to all orders in $g$ one has to consider longer range interactions.

The final thing to check is whether there is an $R$-matrix which solves the Yang-Baxter equation and makes $\mathcal{L}$ satisfy the fundamental commutation relations (3). If we fix the normalisation of $R$ such that its $(1,1)$-component is 1, then (3) gives a unique solution. The form of $R$ is not very enlightening, but it can still be factorised into four terms

$$\check{R}_{12,34}(u, v) = \check{\mathcal{L}}_{234}(u)\check{\mathcal{L}}_{123}(u-v)\tilde{R}_{34}\hat{R}_{23}, \qquad (62)$$

where now $\tilde{R}$ and $\hat{R}$ contain range 3 terms, *i.e.*

$$\tilde{R}_{i,i+1} = 1 + \alpha_i P_{i,i+1} + \beta_i P_{i+1,i+2} + \gamma_i P_{i-1,i} + \delta_i P_{i,i+2} + \epsilon_i P_{i-1,i+1}. \qquad (63)$$

It is easy to check that this $R$-matrix satisfied the Yang-Baxter equation to order $g^2$. This $R$-matrix satisfies the Yang-Baxter equation and braiding unitarity. Finally, at $v = 0$ it factorises into a product of Lax operators confirming Conjecture 4 from [14]. The above decomposition is not really unique. There are several ways of decomposing $R$ in four factors and the factors also depend on the explicit form of the Lax operator.

# 6 Long-range deformations of 6-vertex models

The long-range deformations of 6-vertex models have been studied in [15] and they were classified. Here we now take a different approach, we will classify them by finding the Yang-Baxter integrable ones. One important question is whether the two sets coincide or if the YB integrable ones are a subset.

To demonstrate our method in a well-known setting, let us work out the general procedure for a Hamiltonian of 6-vertex type. We assume a constant Hamiltonian of the form

$$\mathcal{H} = h_1 + h_2(\sigma_z \otimes 1 - 1 \otimes \sigma_z) + h_3\sigma_+ \otimes \sigma_- + h_4\sigma_- \otimes \sigma_+ + h_5(\sigma_z \otimes 1 + 1 \otimes \sigma_z) + h_6\sigma_z \otimes \sigma_z. \tag{64}$$

Following the classification in [3] we know which $R$-matrices are allowed and that there are two independent cases corresponding to $h_6 \neq 0$ and $h_6 = 0$, which we called 6-vertex A and B, respectively. In the case where $h_6 \neq 0$, we found the following Lax operator[2]

$$\mathcal{L}(u) = e^{(h_1 + 2h_5 + h_6)u} \begin{pmatrix} 1 & 0 & 0 & 0 \\ 0 & \frac{h_4 e^{-4h_5 u}}{2h_6 + \omega \coth u\omega} & \frac{\omega e^{-2(h_2 + h_5)u}}{2h_6 \sinh u\omega + \omega \cosh u\omega} & 0 \\ 0 & \frac{\omega e^{2(h_2 - h_5)u}}{2h_6 \sinh u\omega + \omega \cosh u\omega} & \frac{h_3}{2h_6 + \omega \coth u\omega} & 0 \\ 0 & 0 & 0 & e^{-4h_5 u} \end{pmatrix}, \tag{65}$$

where[3] $\omega = \sqrt{4h_6^2 - h_3 h_4}$. Notice that $h_1$ corresponds to an overall normalisation of $\mathcal{L}$ and $h_2$ to a local basis transformation in the auxiliary space. We will use this to put $h_1 = -h_6$ and $h_2 = 0$ in the remainder of this section.

We find that this Lax operator satisfies the fundamental commutation relations with the $R$-matrix.

$$R(u,v) = \begin{pmatrix} 1 & 0 & 0 & 0 \\ 0 & \frac{h_4 e^{-4h_5 u}}{2h_6 + \omega \coth(u-v)} & \frac{\omega e^{-2(h_2 + h_5)(u-v)}}{2h_6 \sinh \omega(u-v) + \omega \cosh \omega(u-v)} & 0 \\ 0 & \frac{\omega e^{2(h_2 - h_5)(u-v)}}{2h_6 \sinh \omega(u-v) + \omega \cosh \omega(u-v)} & \frac{h_3 e^{4h_5 v}}{2h_6 + \omega \coth \omega(u-v)} & 0 \\ 0 & 0 & 0 & e^{-4h_5(u-v)} \end{pmatrix}. \tag{66}$$

Notice that the expression that we find for $R, \mathcal{L}$ are also valid in the $h_6 \to 0$ limit, where this should reduce to a special case of the 6-vertex B operators.

Let us now add a range 3 operator and hence we consider the Hamiltonian density as

$$\mathcal{H}_{123}^{(r=3)} = \mathcal{H}_{12} + g^2 \tilde{\mathcal{H}}_{123}, \tag{67}$$

where $\mathcal{H}_{12}$ is constructed from (64) and $\tilde{\mathcal{H}}_{123}$ is given by

$$\tilde{\mathcal{H}}_{123} = \sum_{a,b,c} \tilde{h}_{a,b,c} \sigma_1^a \sigma_2^b \sigma_3^c, \quad \text{and} \quad \left[\tilde{\mathcal{H}}_{123}, \sum_{i=1}^{3} \sigma_i^z\right] = 0, \tag{68}$$

---

[2]This case can be mapped to the XXZ model.

[3]In order to simplify the form of the expressions in this paper we are writing in several places $h_3$, $h_4$ and $\omega$. Notice however that only two of those are independent and when doing our procedure it is important to choose two and consistently use them all the time.

where the sum runs over $a = 0, \pm, z$. Consider the Lax operator

$$\mathcal{L}_{123}^{(r=3)}(u) = P_{23}P_{12}\left(P_{12}\mathcal{L}_{12}(u) + g^2 \sum_{i\geq 1} \frac{\tilde{\mathcal{L}}_{123}^{(i)} u^i}{i!}\right), \tag{69}$$

where $\mathcal{L}_{12}(u)$ is given by the matrix (31). The $g^0$ part is again only embedded in spaces 1 and 2, just as in the $SU(2)$ sector from the previous section. In this way the boundary conditions at leading order in $g$ are automatically satisfied.

The matrix $\tilde{\mathcal{L}}_{123}^{(i)}$ should conserve total spin and takes the form

$$\tilde{\mathcal{L}}_{123}^{(i)} = \begin{pmatrix} \tilde{l}_{11}^{(i)} & 0 & 0 & 0 & 0 & 0 & 0 & 0 \\ 0 & \tilde{l}_{22}^{(i)} & \tilde{l}_{23}^{(i)} & 0 & \tilde{l}_{25}^{(i)} & 0 & 0 & 0 \\ 0 & \tilde{l}_{32}^{(i)} & \tilde{l}_{33}^{(i)} & 0 & \tilde{l}_{35}^{(i)} & 0 & 0 & 0 \\ 0 & 0 & 0 & \tilde{l}_{44}^{(i)} & 0 & \tilde{l}_{46}^{(i)} & \tilde{l}_{47}^{(i)} & 0 \\ 0 & \tilde{l}_{52}^{(i)} & \tilde{l}_{53}^{(i)} & 0 & \tilde{l}_{55}^{(i)} & 0 & 0 & 0 \\ 0 & 0 & 0 & \tilde{l}_{64}^{(i)} & 0 & \tilde{l}_{66}^{(i)} & \tilde{l}_{67}^{(i)} & 0 \\ 0 & 0 & 0 & \tilde{l}_{74}^{(i)} & 0 & \tilde{l}_{76}^{(i)} & \tilde{l}_{77}^{(i)} & 0 \\ 0 & 0 & 0 & 0 & 0 & 0 & 0 & \tilde{l}_{88}^{(i)} \end{pmatrix}. \tag{70}$$

Sometimes it will be useful to write the whole sum instead of the perturbative formula. For this reason we also define $\lambda_{ij} = \sum_{k\geq 1} \frac{\tilde{l}_{ij}^{(k)} u^k}{k!}$.

We then apply a combined set of strategies to discover which type of deformation is allowed and which kind of freedom do we have. We first follow a very similar procedure to the one discussed in section 3, but just at order $u^2$. With this we discover that five other conditions are required on the elements of $\tilde{\mathcal{H}}_{123}$ in order for the deformation to be integrable, namely

$$\tilde{h}_{-z+} = -\frac{h_4^2}{h_3^2}\tilde{h}_{+z-}, \qquad\qquad \tilde{h}_{z0z} = \frac{h_4}{4h_3}\tilde{h}_{+0-} + \frac{h_3}{4h_4}\tilde{h}_{-0+}, \tag{71}$$

$$\tilde{h}_{z+-} = -\tilde{h}_{+-z} - \frac{4h_6}{h_3}\tilde{h}_{+z-}, \qquad\qquad \tilde{h}_{z-+} = \frac{4h_4h_6}{h_3^2}\tilde{h}_{+z-} - \tilde{h}_{-+z}, \tag{72}$$

$$\tilde{h}_{zzz} = 0. \tag{73}$$

Notice that this gives us a total of five relations on ten possible range 3 deformations. This matches exactly with the long-range deformations found in [15]. As is easy to check, we have the same operators that are allowed.

The next step we performed was to compute the Hamiltonian $\mathbb{H}$ and the transfer matrix $t(u)$ for a periodic spin chain with $L = 6$ sites and solve $[t(u), \mathbb{H}] = 0$ for the coefficients in the Lax, with this we found the solution given in Appendix A. As noted before, there is residual freedom and this needs to be partially fixed by considering the boundary conditions. Writing

$$\lambda_{ij} = A_{ij}u + B_{ij}u^2 + \dots \tag{74}$$

Then we find that

$$A_{32} = \frac{4h_4h_6}{h_3^2}\tilde{h}_{+z-} - \tilde{h}_{-+z}, \qquad A_{33} = A_{66} = -A_{44} = -A_{55} = \frac{h_4}{4h_3}\tilde{h}_{+0-} + \frac{h_3}{4h_4}\tilde{h}_{-0+},$$

$$A_{46} = -\tilde{h}_{+-z}, \qquad\qquad A_{53} = \tilde{h}_{-+z}, \tag{75}$$

$$A_{74} = \frac{h_4^2}{h_3^2}\tilde{h}_{+z-} + \tilde{h}_{-0+}, \qquad A_{25} = \tilde{h}_{+0-} + \tilde{h}_{+z-},$$

and

$$B_{74} = \frac{h_4^2 \tilde{h}_{+0-}(5\omega^2 - 24(2h_5^2 + h_6^2))}{48h_3^2 h_5} - \frac{h_4(h_3^2 B_{5,3} + \omega^2 B_{4,6} - 4h_6^2 B_{4,6})}{8h_3^2 h_5} - \frac{4h_5 h_4^2 \tilde{h}_{+z-}}{h_3^2}$$
$$+ \frac{\tilde{h}_{-0+}(24(h_6^2 - 6h_5^2) - 5\omega^2)}{48h_5} + \frac{(3h_5 - 2h_6)h_4^2 \tilde{h}_{+-z}}{4h_3 h_5} + \frac{(h_5 - 2h_6)h_4 \tilde{h}_{-+z}}{4h_5}, \tag{76}$$

$$B_{25} = \frac{4h_6^2 B_{4,6} - h_3^2 B_{5,3} - \omega^2 B_{4,6}}{8h_4 h_5} + \frac{h_3^2 \tilde{h}_{-0+}(24(2h_5^2 + h_6^2) - 5\omega^2)}{48h_4^2 h_5} + 4h_5 \tilde{h}_{+z-}$$
$$+ \frac{\tilde{h}_{+0-}(144h_5^2 - 24h_6^2 + 5\omega^2)}{48h_5} - \frac{(3h_5 + 2h_6)h_3^2 \tilde{h}_{-+z}}{4h_4 h_5} - \frac{(h_5 + 2h_6)h_3 \tilde{h}_{+-z}}{4h_5}. \tag{77}$$

For this Lax operator we then compute the $R$-matrix and check that it satisfies the Yang–Baxter equation. *Hence all long-range deformations obtained from the deformation equation are Yang-Baxter integrable for 6-vertex models.*

It is interesting to highlight that until range 3, the limit $h_6 \to 0$ is well defined and we can therefore take this limit directly in our Lax operator and R-matrix.

# 7 General formalism and higher range

Let us now extend this to range four and see if we can recognise a pattern. This would correspond to the three-loop Dilation operator in the $SU(2)$ sector. As remarked in [9], in this case integrability does not fix the form of the operator and extra input is needed.

**Range 4** It is easy to check that there are 6 independent operators of range 4 that are invariant under $SU(2)$, which are

$$\rho_1 = A_1 \sum_{i=1}^{3} \sigma_i \otimes 1 \otimes 1 \otimes \sigma_i, \qquad \rho_2 = A_2 \sum_{i,j=1}^{3} \sigma_i \otimes \sigma_j \otimes \sigma_j \otimes \sigma_i, \tag{78}$$

$$\rho_3 = A_3 \sum_{i=1}^{3} \sigma_i \otimes \sigma_j \otimes \sigma_i \otimes \sigma_j, \qquad \rho_4 = A_4 \sum_{i=1}^{3} \sigma_i \otimes \sigma_i \otimes \sigma_j \otimes \sigma_j, \tag{79}$$

$$\rho_5 = A_5 \, \epsilon^{ijk} \sigma_i \otimes \sigma_j \otimes 1 \otimes \sigma_k, \qquad \rho_6 = A_6 \, \epsilon^{ijk} \sigma_i \otimes 1 \otimes \sigma_j \otimes \sigma_k. \tag{80}$$

When discussing the range 3 deformation, we saw that this was unique up to a trivial deformation with $\mathbb{Q}_3$ if we impose $SU(2)$ symmetry. Hence, we will look for a deformation of the form

$$\mathcal{H}_{1234} = \mathcal{H}_{123} + g^4 \sum_i \rho_i, \tag{81}$$

where $\mathcal{H}_{123}$ is given by (51). We can then compute the Lax operator and we notice a few things. First, not all terms are integrable. In particular, we find that (81) is only integrable if

$$A_3 = \frac{1 - 2A_1 - 4A_2}{4}, \qquad A_4 = \frac{1 - 2A_1}{4}. \tag{82}$$

Second, we note that if we impose that our Hamiltonian is parity invariant (or symmetric), then $A_5 = A_6 = 0$ and we are left with just two possible integrable deformations. One of them will be $\mathbb{Q}_4$ from the original XXX spin chain. Hence also at this level, there is only one non-trivial integrable deformation of our Hamiltonian. The relative coefficients, however, between the different terms are not fixed.

We also notice something interesting. The presence of the range 3 part automatically implies that we need to have a non-zero range 4 part. Indeed, we can not set all $A_i = 0$ due to the conditions (82). This agrees with the fact that we could not extend our range 3 deformation to order $g^4$.

The explicit form of $\mathcal{L}$ is not completely fixed and depends on some free functions. We present its explicit form in Appendix B. The explicit form of the R-matrix was not explicitly written here because it is very long. But it can be easily computed perturbatively for up to three loops by plugging the Lax operators up to this order in the fundamental commutation relations. As mentioned before this equation is linear in the R-matrix and therefore simple to solve on Mathematica. We further checked that it perturbatively satisfies the Yang-Baxter equation up to order $g^4$.

**Higher range**   We can now very efficiently compute the Lax operators of perturbative long-range interactions recursively. Let us define $\check{\mathcal{L}}$ as

$$\mathcal{L}_{a_1,\ldots,a_n,i} = P_{a_1 i} \ldots P_{a_n i} \check{\mathcal{L}}_{a_1,\ldots,a_n,i} \,. \tag{83}$$

Let us denote the different possible terms of range $r$ by $A_r^{(i)}$, then we want

$$\mathcal{H}_r = \mathcal{H}_{r-1} + g^{2r} \sum_i c_i A_r^{(i)} \,, \tag{84}$$

for some constants $c_i$. We can then make the following Ansatz for our Lax operator

$$\check{\mathcal{L}}_{a_1,\ldots,a_n,i}(u) = \check{\mathcal{L}}_{a_1,\ldots,a_n}(u) + g^{2n} \mathcal{A}_{a_1,\ldots,a_n,i} \,. \tag{85}$$

The first part in the above expression guarantees that this Lax operator will satisfy the correct boundary conditions up to order $g^{2n-2}$, since it is the lower-order Lax operator embedded in the first $n$ coordinates.

The new information is encoded in $\mathcal{A}$ and since it is at the highest order in $g$, we can actually derive a set of linear equations for it. In particular, we can compute

$$[T^{(g^{2n})}, \mathbb{H}] = 0 \,, \tag{86}$$

on a spin chain of sufficient length. This equation is automatically satisfied up to order $g^{2n}$ and at this order it gives a set of linear equations for $\mathcal{A}$, which are straightforward to solve. Similarly, from the RLL relations we obtain a linear set of equations for the $R$-matrix. However, we were unfortunately not able to find an elegant recursive structure on the $R$-matrix. Nevertheless, we found that Ansatz (85) works.

**Deformation equation**   The remaining question is how the deformation equation factors in this story. We have demonstrated a one-to-one correspondence between the known long-range deformations and integrable Lax operators. Let us now work out at first order what happens in the deformation equation. Let us write

$$\check{\mathcal{L}}_{a_1,a_2,i}(u) = \check{\mathcal{L}}_{a_1,a_2}(u) + g^2 \sum_{n>0} \mathcal{A}_{a_1,a_2,i}^{(n)} u^n \,. \tag{87}$$

Then we can easily find that to order $g^2$

$$\mathcal{H}_{abc} = \mathcal{H}_{ab} + g^2 \mathcal{A}_{abc}^{(1)} \,, \tag{88}$$

$$(\mathcal{Q}_3)_{abcd} = [\mathcal{H}_{ab}, \mathcal{H}_{bc}] + \dddot{\mathcal{L}}_{ab} + g^2 \big([\mathcal{A}_{abc}^{(1)}, \mathcal{H}_{bc} + \mathcal{H}_{cd}] + [\mathcal{H}_{ab}, \mathcal{A}_{bcd}^{(1)}] + \mathcal{A}_{abc}^{(2)}\big) \,. \tag{89}$$

Let us now plug this into the deformation equation (44). In particular, the $g^2$ part of the above charges can be written as a commutator with $X$. Hence, when we apply it to $\mathcal{H}$ we find

$$\sum_a \mathcal{A}^{(1)}_{a,a+1,a+2} = \sum_a [X, \mathcal{H}_{a,a+1}]. \tag{90}$$

At next order, however, things become more interesting. We find

$$\sum_a [\mathcal{A}^{(1)}_{a-1,a,a+1}, \mathcal{H}_{a,a+1} + \mathcal{H}_{a+1,a+2}] + [\mathcal{H}_{a-1,a}, \mathcal{A}^{(1)}_{a,a+1,a+2}] + \mathcal{A}^{(2)}_{a-1,a,a+1}$$

$$= \sum_a [X, [\mathcal{H}_{a-1,a}, \mathcal{H}_{a,a+1}] + \ddot{\mathcal{L}}_{a-1,a}]. \tag{91}$$

By using the result of the NN term, we can rewrite this as

$$\mathcal{A}^{(2)}_{a-1,a,a+1} = \sum_a [X, \ddot{\mathcal{L}}_{a-1,a}] - [[X, \mathcal{H}_{a-1,a}], \mathcal{H}_{a+1,a+2}]. \tag{92}$$

The right hand side is *a priori* of range 4 while the left hand side is of range 3. In general we see that the deformation equation will involve commutators of increasing range and it is not clear that those can be written in terms of a term of range 3. We see that at best the Lax operator can only be implicitly defined perturbatively by the deformation equation. It would be very interesting to see if a deformation equation can be written down for the full Lax operator.

# 8 Possible applications

Finally, let us highlight some possible applications of our results. There clearly are applications in $\mathcal{N} = 4$ SYM theory and related theories. The spectrum for the integrable theories in the AdS/CFT correspondence is mainly solved, but the question of form factors and correlation functions is still open. Many of these approaches use the explicit wave function of the operators at higher loops, see *e.g.* [19–21]. We have constructed the explicit Lax operator and the corresponding $R$-matrix at two-loops and that should open the door for applications of the algebraic Bethe Ansatz for loops and construct the eigenstates with the creation operators corresponding to these operators.

**Correlators** Second, there is an elegant application of this to computing form factors and correlation functions on spin chains in general. This was used in [22] to compute certain correlation functions of the XXZ spin chain.

Consider an integrable spin chain that depends continuously on some parameter $\epsilon$. Let us denote the conserved charges by $\hat{\mathbb{Q}}^{\epsilon}_i$. The operator $\hat{\mathbb{Q}}^{\epsilon}_i$ generically has interaction range $i$. This means that $\hat{\mathbb{Q}}^{\epsilon}_2$ corresponds to a term with nearest neighbor interaction and is usually taken to be the Hamiltonian of the spin chain.

Assume that we can diagonalize these conserved quantities by means of a Bethe Ansatz. Let $|\mathbf{u}^{\epsilon}\rangle$ be an eigenstate of all the operators $\hat{\mathbb{Q}}^{\epsilon}_i$ with corresponding eigenvalues $Q^{\epsilon}_i$, *i.e.*

$$\hat{\mathbb{Q}}^{\epsilon}_i |\mathbf{u}^{\epsilon}\rangle = Q^{\epsilon}_i |\mathbf{u}^{\epsilon}\rangle = \left[ \mathfrak{q}^{\epsilon}_i + \sum_n q^{\epsilon}_i(u^{\epsilon}_n) \right] |\mathbf{u}^{\epsilon}\rangle. \tag{93}$$

We have split the eigenvalue $Q$ into a constant term $\mathfrak{q}$ and a magnon contribution given by the dispersion relation $q^{\epsilon}_i(u^{\epsilon}_n)$.

Next, we interpret $\epsilon$ as a small parameter around which we do quantum mechanical perturbation theory. We expand the operators and their eigenvalues as power series in $\epsilon$ as

$$\hat{\mathbb{Q}}^{\epsilon}_i = \hat{\mathbb{Q}}_i + \epsilon\, \hat{\mathbb{Q}}'_i + \dots, \qquad\qquad Q^{\epsilon}_i = Q_i + \epsilon\, Q'_i + \dots \tag{94}$$

The first correction in $\epsilon$ to the eigenvalue $Q_i^\epsilon$ is given by the expectation value of $\hat{\mathbb{Q}}_i'$ in the unperturbed system

$$Q_i' = \langle \mathbf{u}^0 | \hat{\mathbb{Q}}_i' | \mathbf{u}^0 \rangle \,. \tag{95}$$

Since we have a closed formula for $Q_i'$, we can derive an exact formula for $\langle \mathbf{u}^0 | \hat{\mathbb{Q}}_i' | \mathbf{u}^0 \rangle$. More precisely, from (93), we derive

$$\langle \mathbf{u}^0 | \hat{\mathbb{Q}}_i' | \mathbf{u}^0 \rangle = \mathfrak{q}_i' + \sum_n \left[ \frac{\partial q_i^\epsilon}{\partial \epsilon} + \frac{\partial q_i^\epsilon}{\partial u_n^\epsilon} \frac{\partial u_n^\epsilon}{\partial \epsilon} \right]\Bigg|_{\epsilon=0} \,. \tag{96}$$

Apart from $\frac{\partial u_n^\epsilon}{\partial \epsilon}$ all the terms in (96) depend on the explicit form of the dispersion relations in the integrable model. The correction to the rapidities can be computed from the Bethe equations.

Suppose the rapidities of the magnons in our spin chain satisfy a set of Bethe equations. Generically, the Bethe equations can be written in the form

$$\Phi_i = m_i \in \mathbb{Z} \,, \tag{97}$$

where $\Phi$ is the counting function. We can then express $\frac{\partial u^\epsilon}{\partial \epsilon}$ in terms of the usual Gaudin matrix $G_{ij} = \partial_{u_i} \Phi_j$ as follows

$$\langle \mathbf{u}^0 | \hat{\mathbb{Q}}_i' | \mathbf{u}^0 \rangle = \mathfrak{q}_i' + \frac{\partial q_i^\epsilon}{\partial u_n^\epsilon} \cdot G_{nm}^{-1} \cdot \frac{\partial \Phi_m}{\partial \epsilon} + \sum_n \frac{\partial q_i^\epsilon(u_n)}{\partial \epsilon}\Bigg|_{\epsilon=0} \,. \tag{98}$$

All the quantities in the above formula are known and it allows us to derive closed formulas for a large class of operators in a large class of integrable models.

This means that for instance in any 6-vertex model (64) we can exactly compute the expectation values of any two-site operator that preserves spin. However, it also means that, once the Bethe Ansatz for the long-range model has been carried out, that the expectation values of the different types of operators of the long-range deformations can also be computed analytically. This will provide a large class of correlations functions that can be computed explicitly.

# 9 Conclusions and discussion

In this paper we have demonstrated how to check if a Hamiltonian descends from a Lax operator satisfying the RLL relations. We have applied our approach on the known 6-vertex cases and found agreement with our earlier classification [3].

Subsequently, we applied our method to perturbative long-range deformations of spin chains [12]. These types of deformations naturally appear in integrable models that arise in AdS/CFT. We found that the known integrable deformations of this type all derive from a Lax operator and $R$-matrix. We present the Lax operator and $R$-matrix for the $SU(2)$ sector in $\mathcal{N} = 4$ SYM up to two loops and the Lax operator up to three loops. We also constructed the Lax operator and $R$-matrix for range 3 deformations of the 6-vertex model. We find some recursive structure that relates the orders.

There are many interesting future directions in which this work could be extended. It would be interesting to understand the explicit Lax operator and corresponding $R$-matrix to all orders. This would include wrapping which is discussed in [16]. One can also use our method to compute perturbatively the R-matrix and the Lax operator for other sectors in $\mathcal{N} = 4$ SYM.

Perturbative long-range interactions also have been formulated for open spin chains [23] and it would be interesting to consider such a system. Maybe there are some other applications to long range spin chains appearing from different approaches and contexts like free-fermions [24–26] and coming from inhomogeneities in the bulk [27, 28] and in the boundary [29]. Finally, now that the ingredients for the algebraic Bethe Ansatz were computed, it would be interesting to see if our approach can also help with the computation of form factors and correlation functions in AdS/CFT.

## Acknowledgments

We would like to thank T. Gombor, C. Paletta, B. Pozsgay and A. Prybitok for discussions. We would like to thank C. Paletta and B. Pozsgay for valuable comments on the manuscript.

**Funding information**    MdL was supported by SFI, the Royal Society and the EPSRC for funding under grants UF160578, RGF\R1\181011, RGF\EA\180167 and 18/EPSRC/3590. A.L.R. is supported by the grant 18/EPSRC/3590.

## A   Range 3 Lax operator for 6-vertex model

In this appendix we present explicitly the range 3 (order $g^2$) Lax operator for the 6 vertex model discussed in section 6.

There exist a few combinations that appear very often in the elements of the Lax, so in order to make the expressions more compact we define

$$f_n^{\pm}(u) = \omega \pm n h_6 \tanh u\omega \,. \tag{A.1}$$

The elements of the Lax are then given by

$$
\begin{aligned}
\lambda_{11}(u) = {} & \frac{e^{2uh_5}\omega}{4h_4^2} \\
& \times \frac{\omega^2 + 4h_6^2 + \cosh(2u\omega)(\omega f_4^-(2u) - 12h_6^2) + \cosh(4u\omega)(-2\omega^2 + 2\omega f_2^+(4u) + 8h_6^2)}{f_2^+(u)f_2^-(u)\cosh^3 u\omega \sinh u\omega} \tilde{h}_{-0+} \\
& - \frac{e^{2uh_5}\omega}{h_3^2} \frac{f_2^+(u)\tilde{h}_{+0-}}{f_2^-(u)\sinh(2u\omega)} + \frac{4e^{2uh_5}h_6}{h_3^2} \frac{\omega f_4^-(u) - 12h_6^2}{f_2^+(u)f_2^-(u)} \tilde{h}_{+z-}\tanh^2 u\omega \\
& + \frac{e^{2uh_5}}{h_3} \frac{f_4^+(u) + \omega \tanh^2 u\omega}{f_2^+(u)} \tilde{h}_{+-z} + \frac{e^{-2uh_5}f_2^+(u)^2 \coth u\omega}{2h_3^2 \sinh u\omega}\lambda_{25}(u) \\
& + \frac{e^{2uh_5}}{h_4} \frac{\omega^2 - 8h_6^2 + 2h_6(4h_6\cosh(2u\omega) + 2\omega\sinh(2u\omega) - 3\omega\tanh u\omega)}{f_2^+(u)f_2^-(u)\cosh^2 u\omega}\tilde{h}_{-+z} \\
& + \frac{2e^{2uh_5}h_6 f_2^+(2u)\cosh(2u\omega)}{h_4\omega\cosh u\omega}\lambda_{32}(u) + \frac{e^{2uh_5}f_2^+(u)\cosh u\omega}{\omega}\lambda_{33}(u) \\
& - \frac{e^{2uh_5}f_2^+(2u)\cosh(2u\omega)}{2\omega\cosh u\omega}\lambda_{44}(u) + \frac{f_4^+(u)\coth u\omega + \omega\tanh u\omega}{2h_3}\lambda_{46}(u) \\
& - \frac{e^{4uh_5}\omega}{h_4\sinh(2u\omega)}\lambda_{53}(u) + \frac{e^{2uh_5}}{2\cosh u\omega}\lambda_{55}(u) - \frac{e^{6uh_5}}{4\omega h_4^2}\left(\frac{\omega^3 + \omega\cosh(2u\omega)(\omega^2 - 12h_6^2)}{\sinh^2 u\omega \cosh u\omega}\right.
\end{aligned}
$$

$$+\frac{2h_6(2h_6(\omega+2\omega\cosh(4u\omega)+16h_6\cosh u\omega\sinh^3 u\omega)+\omega^2\sinh(4u\omega))}{\sinh^2 u\omega\cosh u\omega}\Bigg)\lambda_{74}(u),$$

$$(A.2)$$

$$\lambda_{22}(u)=\lambda_{11}(u)-e^{2uh_5}\left(\frac{h_4}{h_3}\frac{\tilde{h}_{+0-}\tanh u\omega}{f_2^-(u)}+\frac{h_3}{h_4}\frac{\tilde{h}_{-0+}\tanh u\omega}{f_2^+(u)}+4\frac{h_4h_6}{h_3}\frac{\tilde{h}_{+z-}\tanh^2 u\omega}{f_2^+(u)f_2^-(u)}\right.$$

$$\left.-\frac{f_2^-(2u)\cosh 2u\omega+\omega}{f_2^+(u)f_2^-(u)^2}\frac{\tanh^2 u\omega}{\cosh^2 u\omega}h_3\tilde{h}_{-+z}+\frac{f_2^+(u)\cosh u\omega}{\omega}(\lambda_{55}(u)-\lambda_{66}(u))\right),$$

$$(A.3)$$

$$\lambda_{23}(u)=e^{2uh_5}\left(-\frac{\omega^2}{h_3}\frac{\tilde{h}_{+0-}\text{sech}^2 u\omega}{f_2^+(u)f_2^-(u)}-\frac{4h_6}{h_3}\frac{\tilde{h}_{+z-}\tanh u\omega}{f_2^-(u)}+\frac{2h_3}{h_4}\frac{\tilde{h}_{-+z}\tanh u\omega}{f_2^-(u)}\right.$$

$$\left.+\frac{\omega h_3}{h_4^2}\frac{f_2^+(2u)\tilde{h}_{-0+}(1+\tanh^2 u\omega)}{f_2^+(u)f_2^-(u)}\right)+\frac{e^{-2uh_5}\omega}{h_3\sinh u\omega}\lambda_{25}(u)$$

$$+\frac{e^{2uh_5}h_3f_2^+(u)}{\omega h_4}\cosh u\omega\,\lambda_{32}(u)-\frac{e^{6uh_5}h_3f_2^+(u)^2}{\omega h_4^2}\frac{\cosh^2 u\omega}{\sinh u\omega}\lambda_{74}(u),$$

$$(A.4)$$

$$\lambda_{35}(u)=e^{2uh_5}\left(\frac{2\tilde{h}_{+z}\tanh u\omega}{f_2^+(u)}+\frac{4h_6}{h_3}\frac{f_6^-(u)\tilde{h}_{+z-}\tanh u\omega}{f_2^+(u)f_2^-(u)}-\frac{h_4\tilde{h}_{+0-}\tanh^2 u\omega}{f_2^+(u)f_2^-(u)}\right.$$

$$+\frac{h_3}{h_4^2}\frac{\omega^2+4h_6\coth u\omega f_1^+(u)}{f_2^+(u)f_2^-(u)}\tilde{h}_{-0+}\tanh^2 u\omega+\frac{8h_3h_6}{h_4}\frac{\tilde{h}_{-+z}\tanh^2 u\omega}{f_2^+(u)f_2^-(u)}$$

$$\left.+\frac{h_3}{\omega}\left(\frac{4h_6}{h_4}\lambda_{32}(u)+\lambda_{33}(u)-\lambda_{44}(u)\right)\sinh u\omega\right)$$

$$+\lambda_{46}-\frac{4e^{6uh_5}h_3h_6}{\omega h_4^2}f_2^+(u)\cosh u\omega\,\lambda_{74}(u),$$

$$(A.5)$$

$$\lambda_{47}(u)=e^{2uh_5}\left(\left(\frac{\tilde{h}_{+0-}\omega}{f_2^+(u)}+\frac{h_3^2}{h_4^2}\frac{\tilde{h}_{-0+}}{f_2^-(u)}\right)\tanh u\omega-2\left(2h_6\tilde{h}_{+z-}-\frac{h_3^2}{h_4}\right)\frac{\tanh^2 u\omega}{f_2^+(u)f_2^-(u)}\right)$$

$$+\frac{e^{2uh_5}h_3^2}{\omega h_4}\sinh u\omega\,\lambda_{32}(u)-\frac{e^{6uh_5}h_3^2}{\omega h_4^2}f_2^+(u)\cosh u\omega\,\lambda_{74}(u),$$

$$(A.6)$$

$$\lambda_{52}(u)=e^{-2uh_5}\left(\frac{2\omega\tilde{h}_{-0+}\tanh u\omega}{f_2^+(u)f_2^-(u)}-2h_4\left(\frac{2h_4h_6}{h_3^2}\tilde{h}_{+z-}-\tilde{h}_{-+z}\right)\frac{\tanh^2 u\omega}{f_2^+(u)f_2^-(u)}\right)$$

$$+\frac{e^{-2uh_5}h_4}{\omega}\sinh u\omega\,\lambda_{32}(u)-\frac{e^{2uh_5}}{\omega}f_2^+(u)\cosh u\omega\,\lambda_{74}(u),$$

$$(A.7)$$

$$\lambda_{64}(u)=\frac{e^{-4uh_5}h_4}{h_3}\lambda_{35}(u)-\frac{2e^{-2uh_5}}{f_2^+(u)}\left(\frac{h_4\tilde{h}_{+-z}}{h_3}+\tilde{h}_{-+z}\right)\tanh u\omega$$

$$-\frac{e^{-2uh_5}h_4}{\omega}\left(\lambda_{33}(u)-\lambda_{44}(u)\right)\sinh u\omega-\frac{e^{-4uh_5}h_4}{h_3}\lambda_{46}(u)+\lambda_{53}(u)$$

$$-\frac{e^{-2uh_5}h_4\sinh u\omega}{\omega}(\lambda_{55}(u)-\lambda_{56}(u)),$$

$$(A.8)$$

$$\lambda_{67}(u) = \frac{\omega e^{2uh_5}h_3}{\sinh u\omega\, h_4^2}\lambda_{52}(u) - \omega\, h_3\left(\frac{\tilde{h}_{+0-}}{h_3^2 f_2^-(u)} + \frac{\tilde{h}_{-0+}}{h_4^2 f_2^+(u)}\right)\mathrm{sech}\,u\omega$$

$$+ 2\omega\left(\tilde{h}_{+-z} + \frac{2h_6\tilde{h}_{+z-}}{h_3}\right)\frac{\mathrm{sech}\,u\omega\,\tanh u\omega}{f_2^+(u)f_2^-(u)} + \frac{e^{-4uh_5}}{h_3}f_2^+(u)\coth u\omega\,\lambda_{25}(u)\,, \quad \text{(A.9)}$$

$$\lambda_{76}(u) = \lambda_{64}(u) - \frac{4e^{-2uh_5}}{f_2^+(u)}\left(\frac{2h_4 h_6\tilde{h}_{+z-}}{h_3^2} - \tilde{h}_{-+z}\right)\tanh u\omega$$

$$+ \frac{e^{-2uh_5}}{\omega}f_2^-(u)\cosh u\omega\,\lambda_{32}(u) + \lambda_{53}(u)$$

$$+ \frac{e^{-2uh_5}}{\omega}\big(h_4\lambda_{55}(u) - h_4\lambda_{66}(u) + h_3\lambda_{74}(u)\big)\sinh u\omega\,, \quad \text{(A.10)}$$

$$\lambda_{77}(u) = e^{-2uh_5}\left(\frac{f_2^+(u)f_2^-(2u)\tilde{h}_{+0-}\coth(2u\omega)}{h_3^2 f_2^-(u)} + \frac{f_2^+(u)(-2\omega + f_2^+(2u)\cosh(2u\omega))\tilde{h}_{-0+}}{f_2^-(u)\sinh(2u\omega)}\right.$$

$$+ \frac{8h_6}{h_3^2}\frac{(\omega f_2^-(u) - 8h_6^2)\tilde{h}_{+z-}\tanh^2 u\omega}{f_2^+(u)f_2^-(u)} - \frac{f_6^-(u)(f_4^+(u) + \omega\tanh^2 u\omega)\tilde{h}_{-+z}}{h_4 f_2^+(u)f_2^-(u)}$$

$$\left.+ \frac{\left(-\omega^2 + 2\omega h_6\mathrm{sech}^2 u\omega\tanh u\omega + (\omega^2 - 8h_6^2)\tanh^2 u\omega\right)\tilde{h}_{+-z}}{h_3 f_2^+(u)f_2^-(u)}\right)$$

$$- \frac{e^{-6uh_5}f_2^+(u)f_2^-(u)\coth u\omega}{2h_3^2\sinh u\omega}\lambda_{25}(u) + \frac{2e^{-2uh_5}h_6 f_2^+(2u)\cosh(2u\omega)}{\omega h_4\cosh u\omega}\lambda_{32}(u)$$

$$+ \frac{e^{-2uh_5}f_2^+(u)\cosh u\omega}{\omega}\big(\lambda_{33}(u) + \lambda_{66}(u)\big) - \frac{e^{-4uh_5}\omega}{h_3\sinh(2u\omega)}\lambda_{46}(u)$$

$$- \frac{e^{-2uh_5}f_2^+(2u)\cosh(2u\omega)}{2\omega\cosh u\omega}\big(\lambda_{44}(u) + \lambda_{55}(u)\big) + \frac{f_4^+(u)\coth u\omega + \omega\tanh u\omega}{2h_4}\lambda_{53}(u)$$

$$+ \frac{e^{2uh_5}}{2\omega h_4^2}\left(\frac{\omega^3\coth u\omega}{\sinh^2 u\omega} - 8h_6\big(4h_6 f_1^+(2u)\coth(2u\omega) + \omega(\omega + h_6\mathrm{csch}(2u\omega))\big)\right)\sinh u\omega\,\lambda_{74}(u)\,,$$

$$\text{(A.11)}$$

and

$$\lambda_{88}(u) = \lambda_{77}(u) + e^{-2uh_5}\left(\frac{h_4\tilde{h}_{+0-}}{h_3 f_2^+(u)} + \frac{h_3\tilde{h}_{-0+}}{h_4 f_2^-(u)}\right)\tanh u\omega$$

$$+ \frac{2e^{-2uh_5}h_4}{f_2^+(u)f_2^-(u)}\left(\tilde{h}_{+-z} + \frac{2h_6}{h_3}\tilde{h}_{+z-}\right)\tanh^2 u\omega$$

$$- \frac{e^{-2uh_5}f_2^+(u)}{\omega}\big(\lambda_{33}(u) - \lambda_{44}(u)\big)\cosh u\omega\,. \quad \text{(A.12)}$$

As one can see above, there is still a lot of freedom in the Lax since everything is written in terms of $\lambda_{25}(u), \lambda_{32}(u), \lambda_{33}(u), \lambda_{44}(u), \lambda_{46}(u), \lambda_{53}(u), \lambda_{55}(u), \lambda_{66}(u)$ and $\lambda_{74}(u)$. These objects need however to satisfy some properties (see equations in (76) and (77)) in order to the boundary conditions be satisfied.

The R-matrix was computed explicitly using the RLL relations and it was proved to satisfy the Yang-Baxter equation.

# B  Range 4 Lax operator

Setting $A_5 = A_6 = 0$, we can write for the $g^4$ part

$$
\begin{aligned}
\mathcal{L}^{(g^4)} = {} & \lambda_1 1 + \lambda_2 \sum \sigma^i \otimes \sigma^i \otimes 1 \otimes 1 + \lambda_3 \sum 1 \otimes \sigma^i \otimes \sigma^i \otimes 1 + \lambda_4 \sum 1 \otimes 1 \otimes \sigma^i \otimes \sigma^i \\
& + \lambda_5 \sum \sigma^i \otimes 1 \otimes \sigma^i \otimes 1 + \lambda_6 \sum 1 \otimes \sigma^i \otimes 1 \otimes \sigma^i \\
& + \lambda_7 \epsilon_{ijk} \sigma^i \otimes \sigma^j \otimes \sigma^k \otimes 1 + \lambda_8 \epsilon_{ijk} 1 \otimes \sigma^i \otimes \sigma^j \otimes \sigma^k \\
& + \lambda_9 \sum \sigma^i \otimes 1 \otimes 1 \otimes \sigma^i + \lambda_{10} \sum \sigma^i \otimes \sigma^j \otimes \sigma^j \otimes \sigma^i + \lambda_{11} \sum \sigma^i \otimes \sigma^j \otimes \sigma^i \otimes \sigma^j \\
& + \lambda_{12} \sum \sigma^i \otimes \sigma^i \otimes \sigma^j \otimes \sigma^j + \lambda_{13} \, \epsilon_{ijk} \sigma^i \otimes \sigma^j \otimes 1 \otimes \sigma^k + \lambda_{14} \epsilon_{ijk} \sigma^i \otimes 1 \otimes \sigma^j \otimes \sigma^k \, .
\end{aligned}
\tag{B.1}
$$

We have pulled out an explicit factor of $u^2$ in order to make manifest where all the $\lambda_i$ are functions of $u$ and they satisfy the following relations

$$
\lambda_4 = \frac{(u-1)^3(3u-2)}{2(1-2u)^2 u^2}(2A_1 - 1) + \left(\frac{2}{u} - 1\right)\lambda_{10}\,,
\tag{B.2}
$$

$$
\lambda_6 = \frac{(u-1)^2}{2(2u-1)^2}\left(\frac{4(2u-1)}{u^2}A_1 - 2A_1 + 1\right) + \left(\frac{2}{u} - 1\right)\lambda_9\,,
\tag{B.3}
$$

$$
\lambda_8 = \frac{i(u-1)^3}{(1-2u)^2(u-2)}\left(A_1 - \frac{(2A_1 - 1)(u+2)}{4u}\right) + \left(\frac{2}{u} - 1\right)\lambda_{14}\,,
\tag{B.4}
$$

$$
\begin{aligned}
\lambda_{11} = {} & \frac{A_1(u(u(2u-5)+9)-4)(u-1)^2}{2(1-2u)^2 u^2} + \frac{A_2(u-1)^2}{u(2u-1)} - i\lambda_{14} + \left(\frac{2}{u} - 2\right)\lambda_9 \\
& - \frac{\left(u\left(u\left(u\left(4u^2 - 22u + 31\right) - 17\right) - 4\right) + 4\right)(u-1)^2}{4(u-2)^2 u(2u-1)^3} - \frac{u}{2}\lambda_3 + \left(1 - \frac{u}{2}\right)\lambda_5\,,
\end{aligned}
\tag{B.5}
$$

$$
\lambda_{12} = \frac{(1-2A_1)(u-1)^3}{4(1-2u)^2 u} + \lambda_{10}\,,
\tag{B.6}
$$

$$
\lambda_{13} = \frac{iA_1(3u-2)(u-1)^2}{2(1-2u)^2 u} - \frac{3iu(u-1)^2}{4(1-2u)^2(u-2)} + i\lambda_9\,.
\tag{B.7}
$$

In order to be compatible with the boundary conditions, we also need to impose behaviour for $u \to 0$. Let us write

$$
\lambda_i \sim \frac{a_i}{u} + b_i + \mathcal{O}(u)\,.
\tag{B.8}
$$

Then we need that

$$
a_9 = A_1\,, \qquad a_{10} = \frac{1}{2} - A_1\,, \qquad b_9 = \frac{A_1}{2}\,, \qquad b_{14} = \frac{i(2A_1 + 1)}{8}\,,
\tag{B.9}
$$

and all the remaining constants $a_i, b_i$ need to vanish.

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
