# Peer review of "Lifting integrable models and long-range interactions"

_SciPost Physics, doi:SciPost Phys. 15, 241 (2023)_

## Round 2 · Referee Report · Anonymous (Referee 1) · 2023-5-31

Strengths

1- The paper addresses an interesting technical question in the context of integrable spin chains. 2- It is mostly well written and provides a number of useful examples.

Weaknesses

1- Some definitions could be sharpened or better explained. 2- The usefulness of the results could be addressed in more detail.

Report

The paper at hand discusses the interesting conceptual question in how far integrability formulated via conserved charges can be lifted to Lax operators or R-matrices. This paper should be considered in the context of the letter referred to as [16], which appeared at the same time and discusses similar aspects in a systematic way. The discussed situations and examples are useful to demonstrate that indeed in many cases on can find Lax operators and R-matrices that obey Yang-Baxter relations and correspond to the considered charge operators. The implications of the presented findings are not completely conclusive and the usefulness of the discussed results remains a bit opaque which is reflected in the very short section 8. E.g. it is stated that the results "should open the door for applications of the algebraic Bethe Ansatz" but this is not demonstrated explicitly. In particular, the question of wrapping is not addressed as opposed to reference [16] which is one of the most interesting points in the context of long-range spin chains. Still the result provides useful insight into the algebraic structure of integrable systems and progresses a question that has been on the table since the discovery of AdS/CFT type long-range spin chains. I therefore recommend it for publication after the below points have been addressed.

Requested changes

1- The paper refers to a "constant" Hamiltonian as opposed to a "functional" one. These to terms should be defined explicitly. 2- Below eq. (1.1) it is stated that these spin chains are integrable. This should probably mean perturbatively integrable, which should be briefly explained at this place to avoid confusion. 3- Above (2.19) it is stated that R is the unique solution to the Sutherland equation. Is this obvious? 4- The construction around (2.21)-(2.23) is a bit intransparent. The A in the equation below eq. (2.23) carries no site labels, while one term on the right hand side has these labels. How is that to be understood? Does this A enter into the boost operator in (2.22), which would require a density? If yes, what is the density of H^2 in the definition of A below eq. (2.23)? Please clarify these points. It would also be helpful to give an explicit example for Q3 in eq. (2.23), in particular for the contribution H^(1)_{k,k+1}. 5- Is there a reason to use different fonts for the Q's in e.g. (4.1) and (8.1)?

Typos: * in the first paragraph on page 2 there is a doubling of "in the" * last paragraph on page 2: deformation(s) * last paragraph on page 2: We then go (to) three * eq. (2.2): fullstop -> comma * eq. (2.4): d -> d/du * above eq. (3.3): This differ(s) * below (3.7): It -> it * below (6.6): is spaces -> in spaces * around (7.6): \check L vs \check\cal L * below (7.8): that (the) this * last sentence of sec. 8: can be compute(d)

---

## Round 2 · Referee Report · Anonymous (Referee 2) · 2023-6-10

Strengths

  1. innovative
  2. rigorous

Weaknesses

n/a

Report

This paper addresses the question of whether one can construct a Lax operator and and R-matrix in order to demonstrate that a given spin-chain Hamiltonian is integrable. This is done not trying to match the Hamiltonian to one of the classes which are known to be integrable (by the classification of the same authors and collaborators), but directly embedding the Hamiltonian in the tower of charges and constructing the Lax operator and R-matrix algorithmically starting from small number of sites. The procedure is practically and most efficiently carried out by a computer programme. This method is then extended to lang-range spin-chains, for example of the type appearing in the AdS/CFT correspondence.

The paper is interesting and it brings forward previous work on this line of investigation, and it constitutes progress in determining the complete algebraic structure underlying the AdS_5/CFT_4 integrable system. I definitely recommend it for publication. I only have a few minor comments which the authors might perhaps find useful to consider:

  1. It would be useful to recollect at the beginning what the difference between a "constant" and "functional" Hamiltonian is

  2. The meaning of P_{ia} as the permutation operator is never stated

  3. The connection with the SU(2) sector in N=4 super-Yang-Mills is advertised but never truly made very explicitly. It is not quite clear whether this is something which this method will allow to do in the future, or whether hidden in the formulas of this paper one can already find for instance the higher loop Lax operator and R-matrix for AdS/CFT. For one thing, there is no recap of the N=4 dilatation operator expression (perhaps with a little comment on the original notation of the N=4 literature confronted with the one of this paper) for the ease of comparison. Since this is such a central point of the method, it would be good to have a very clear presentation of this aspect in a completely clear and unequivocal way if possible.

Requested changes

Please see report above

---

## Round 2 · Referee Report · Anonymous (Referee 3) · 2023-6-27

Strengths

1 - The paper addresses an interesting long standing question of long range spin chains.
2 - The paper is well written.

Weaknesses

1 - Some definitions are mathematically not rigorous.

Report

The paper addresses a long-standing problem, namely whether there are R-matrices and Lax-operators for the perturbative long-range spin chains which appear in the AdS/CFT correspondence. A compact definition for the perturbative long-range spin chains on the level of charges has been available since 2008. This definition made it possible to determine the asymptotic spectrum, but the disadvantage is that the charges are defined only for infinite length. The Lax operators of spin chains have proven to be very useful for nearest-neighbor interactions, but their generalization for long-range deformations has not been determined yet. This article tries to fill this gap.

The authors generalized the definitions of [14] for perturbative long range deformations. It gives the perturbative definitions of the Lax-operators and R-matrices for the long range spin chains. The long range deformations of the six vertex models have already been classified on the level of charges and the authors demonstrated that there exist Lax-operators and R-matrices for all integrable deformations of the six vertex models in the first order. They also determined the Lax-operator of the dilatation operators of the SU(2) sector in N=4 SYM in three loops.

The article contains significant advances in an important and persistent problem of the AdS/CFT duality, so I recommend it for publication after the below points have been addressed.

Requested changes

See the attached Requested_changes.pdf

Attachment

---

## Round 3 · Referee Report · Anonymous · 2023-10-28

Strengths

innovative
rigorous

Weaknesses

none

Report

The authors have adequately implemented the changes and have added useful explanations. I recommend the publication

Requested changes

none

---

## Round 3 · Referee Report · Anonymous · 2023-10-29

Report

The provided modifications address all the points I had raised and I consider the paper ready for publication.

---

## Round 3 · Referee Report · Anonymous · 2023-11-17

Report

The authors have made the requested changes. I recommend the publication.

---

## Round 3 · Author Response

We thank the Referees for carefully reading our paper and for their constructive comments.
We believe that we addressed all their comments/questions in the new revised version and that the paper has further improved. We hope that the paper will now be found suitable for publication in SciPost Physics.

Please find below our point-by-point responses to the report. The numbers in the replies correspond to the numbers of the questions/comments by the referees. All equation numbers and footnotes refer to the new version of the paper.

---

## Round 3 · List of Changes

Reply to Referee 1:

1. We addressed this point in the new version after equations (2.4) and (2.16) and additionally in footnote 1.
2. We make it now clear that we refer to perturbative integrability and give a brief definition of what I mean by that.
3. The Sutherland equation, as presented in (2.19) is a system of linear ordinary differential equations on the elements of the Lax matrix $ \mathcal{L}(u) $ with specific boundary conditions. It is also an overdetermined system because for a $ N $-dimensional local Hilbert space while $ \mathcal{L}(u) $ has $ (N^2)^2 $ elements, the Sutherland equation has $ (N^3)^2 $ of them. Assuming that such matrix elements and their derivatives are differentiable (which is the case for all examples in the literature so far), the solution of Sutherland equation is indeed unique.
4. We thank the referee for these questions which allowed us to improve this part of the paper. We had used $ A $ for two operators that meant different things. We now fixed this point by calling one of them $ M $. Furthermore, we now added the correct labels in $ A $ under equation (2.23). We have opted at the moment for not adding an example of $ H^{(1)}_{k,k+1} $. The reasons behind are the following: $ H^{(1)}_{k,k+1} $ is non-zero only for functional Hamiltonians, which are the ones generated by non-difference form R-matrices (i.e. cases where $ R(u,v)\neq R(u-v) $), which are generally more complicated than the difference form cases. So, although it is easy to construct this term in Mathematica, it is not easy to write it in a simple enough way. It is also highly model dependent. With this in mind, we believe that our presentation is clearer in this way.
5. We have corrected this point by putting the Q's in all sections with same font.

Additionally:
A) All typos pointed out by the referee were corrected.
B) About the comment "The implications of the presented findings are not completely conclusive and the usefulness of the discussed results remains a bit opaque which is reflected in the very short section 8. E.g. it is stated that the results "should open the door for applications of the algebraic Bethe Ansatz" but this is not demonstrated explicitly."
Our response: The Algebraic Bethe ansatz is usually based on two ingredients: the Lax operator and the R-matrix. Until our paper those had not been available for the su(2) sector in $\mathcal{N}=4$ SYM except at one-loop order. We are working on the computation of the Bethe ansatz at the moment, but we believe that its explicit computation falls out of the scope of what proposed in this paper. In addition our work opens the door to compute the Lax operator and the R-matrix for other sectors in $\mathcal{N}=4$ SYM, like $ su(1|1) $, for example. We modified the conclusions to address these points.

Reply to Referee 2:

1. We addressed this point in the new version after equations (2.4) and (2.16) and additionally in footnote 1.
2. The definition of P_{ia} was added just after equation 2.5.
3. We have now explicitly written the dilatation operator up to three loops (see new equations (5.1-5.3)) in the notation of reference [9] and commented on the connection with our notation. Please see paragraphs around equations (5.1-5.3). In addition, to clarify the point about higher loop Lax operator and R-matrix for AdS/CFT we modified the paragraph just before the subsection "Higher range" in page 14 by writing it as "The explicit form of $\mathcal{L}$ is not completely fixed and depends on some free functions. We present its explicit form in Appendix B. The explicit form of the R-matrix was not explicitly written here because it is very long. But it can be easily computed perturbatively for up to three loops by plugging the Lax operators up to this order in the fundamental commutation relations. As mentioned before this equation is linear in the R-matrix and therefore simple to solve on Mathematica. We further checked that it perturbatively satisfies the Yang-Baxter equation up to order $ g^4 $."
In addition, we also modified the conclusions to address some of these points.

Reply to Referee 3:

1. The first question of the referee is about the recursive structure of the Lax operator and the R-matrix. Our observations in section 7 were based on our observations and explicit calculations. We found that Ansatz (7.8) is correct and works not only for the models studied in the paper but also in other examples worked out afterwards. However, we were not able to find a nice way to recursively relate the factors of the R-matrix. Take for instance, the easiest example (5.9) which is the R-matrix that includes the NNN term. Its lowest order is given by a product of 4 smaller R-matrices as explained around (5.10). In particular, we have

\begin{equation}
R_{12,34}(u,v) = R_{14}(u,0) R_{13}(u,v) R_{24}(0,0) R_{23}(0,v)
\end{equation}

It is not just a simple factorization, but also a non-trivial choice of spectral parameters. Unfortunately, when we go to higher orders in the coupling constant there seem to be no nice factorization properties and higher orders start to affect lower orders. For instance, for the NNNN term, the lowest order of the $R$-matrix is given by a product of 9 usual $R$-matrices $R_{12}$. This can not be build out of $R_{12,34}$ in an easy way. We agree that it would be important to understand the recursive structures that appear here, but this does not seem to be an easy question and would have to be addressed in future work. We have changed the text in Section 7 to reflect this discussion.

2. We have corrected equations (3.6) and (4.7) as suggested.

Extra: In addition to the points raised by the referees we also corrected a few typos
1. L ->\mathcal{L} in equation (2.5)
2. Corrected the references before equation (5.1)
3. Correct a few language typos

---

## Editorial Decision

published